# Exploring emotional well-being, spiritual, religious and personal beliefs and telomere length in chronic pain patients—A pilot study with cross-sectional design

Linn Rönne-Petersén[1]*, Maria Niemi[2], Harald Walach[3], Catharina Lavebratt[4], Liu L. Yang[4], Björn Gerdle[5], Bijar Ghafouri[5], Torkel Falkenberg[1]†

1 Department of Neurobiology, Division of Nursing, Care sciences and Society, Karolinska Institutet, Stockholm, Sweden, 2 Department of Global Public Health, Karolinska Institutet, Research Group "Epidemiology of Psychiatric Conditions, Substance use and Social Environment", Stockholm, Sweden, 3 Change Health Science Institute, Basel, Switzerland, 4 Department of Molecular Medicine and Surgery, Karolinska Institutet and Center for Molecular Medicine, Stockholm, Sweden, 5 Pain and Rehabilitation Centre and Department of Health, Medicine and Caring Sciences, Linköping University, Linköping, Sweden

† Deceased.
* linn.ronne-petersen@ki.se

**Data Availability Statement:** There are ethical restrictions on sharing our de-identified data set publicly since data contain potentially identifying

## Abstract

Living with chronic pain is associated with substantial suffering and high societal costs. Patient reported outcomes (PROM's) and cellular ageing should be considered in pain management. The aim of this study was to explore correlations of PROM's and cellular ageing (telomere length [TL] and telomerase activity [TA]) amongst patients with chronic non-malignant pain. This was an explorative pilot study with cross-sectional design and recruitment was done at two pain rehabilitation facilities in Sweden, with inpatient setting/integrative care and outpatient setting/multimodal care, respectively. Eighty-four patients were enrolled by referral to pain rehabilitation in Sweden. The main outcome measures collected after admission in addition to TL and TA were the following PROMs: Numerical Rating Scale (NRS), Chronic Pain Acceptance Questionnaire (CPAQ), Hospital Anxiety and Depression Scale (HADS), Five Facets Mindfulness Questionnaire (FFMQ), WHO Quality of Life–Spiritual, Religious and Personal Beliefs (WHOQoL-SRPB) and EuroQol 5 Dimensions (EQ-5D). All the PROM's showed evidence of poor overall health status among the participants. TL correlated negatively with HADS score ($r = -.219$, $p = .047$) and positively with WHOQoL-SRPB ($r = .224$, $p = .052$). TL did not correlate with any of the pain measures. TA correlated positively with pain spread ($r = .222$, $p = .049$). A mediation of the direct effect of spiritual well-being on TL by anxiety and depression could be shown ($b = 0.008$; $p = .045$). The correlations between TL and SRPB and anxiety and depression suggest some importance of emotional and SRPB dimensions in pain management, with implications for cellular aging, which may warrant further study.

**Trial registration**: ClinicalTrials.gov Identifier: NCT02459639.

sensitive patient information. The reasons are both the rules on confidentiality, primarily to protect the research subjects, and the GDPR regulations in Sweden. These restrictions are imposed by the institutional legal body thru lawyer Sofia Hallquist Björkegren; sofia.hallquist@ki.se to which data requests may be sent for researchers who meet the criteria for access to confidential data.

**Funding:** This work was supported by Ekhagastiftelsen (TF), KI Research School in Health Care Sciences (LRP), Gyllenbergsstiftelsen (TF; LRP), Integrative Care Science Center (TF;LRP; MN), Vidarstiftelsen (TF) and Mahle stiftung (TF). The funders had no role in study design, data collection and analysis, decision to publish, or preparation of the manuscript.

**Competing interests:** The authors have declared that no competing interests exist.

# Introduction

This study is an exploration of correlations between baseline parameters obtained from a larger, prospective clinical trial. The study spans over broad fields of knowledge, from psychospirituality to genomics. However, in the context of chronic pain, both of these fields are under-researched, which motivates the present study. In addition, this study has an explicit interest in exploring possible links between these two areas of research.

## Chronic pain as a personal and societal condition

Chronic pain constitutes considerable suffering for those afflicted, and poses a great challenge for the healthcare system with high costs for society [1,2]. Chronic pain has been described as an enigmatic phenomenon associated with substantial negative psychosocial impact and physical ill-health [3–5]. With 19–50% of the European population affected, chronic pain is common, and in the female population the prevalence is at least 10% higher than among males [2,6,7]. The complex health issues that chronic pain patients often present are often misunderstood and misinterpreted by the healthcare system, leading to feelings of stigmatization and repeated health care seeking behaviour [8,9]. Leadely et al [1] conclude that chronic pain is neither sufficiently prioritized in policies nor in budgets. The reasons why such a widespread and incapacitating condition isn't more prioritized seems largely unclear.

## Measuring pain

Despite the complex nature of pain and the subsequent need for a broader approach to its' study, most pain research has mainly focused on pain intensity and duration [10]. In contrast, Wideman et al [10] have proposed a comprehensive model for how to integrate the subjective and objective aspects of pain, while at the same time arguing that the most important component of pain, namely the *pain experience (inner experience)*, cannot be measured since it is non-observable and deeply personal. What research and clinicians can quantify are *pain measures* (subjective self-reported or objective non-self-reported), bearing in mind that *what* is being measured is not pain experience in its' broad sense, but a limited part of the pain phenomena.

## Telomeres, cellular ageing and pain

Telomeres are specific, protective structures at the ends of chromosomes that are crucial for preventing genetic information from being lost during cell division [11]. Shortened telomere length (TL) is considered a predictor for several age-related diseases and mortality [12–14], although it is also affected by factors *independent* of chronological age [14–16]. TL shortening independent of age tends to result from cumulative life stress and age-related diseases [17]. The enzyme telomer*ase* functions as a buffer against cellular aging in the nucleus by steadily re-building the telomeres by adding on base pairs [11,16]. Changes in TL can take months or years to be detectable, while telomerase activity (TA) is more dynamic and linked to short-term changes in health and lifestyle. Changes in TA enzyme levels have implications for telomere length and the protection of chromosomes [18,19], making it a suitable measure for examining more immediate effects of certain lifestyle factors or health-promoting behaviors [20].

Some studies address the association between chronic pain and TL, whereas reports on pain and measures of TA are lacking. The majority of studies find significant correlations between chronic pain or degenerative tissue (associated with pain) and decreased TL [5,21–24]. However, some other studies have not found any correlations between the two [25,26].

Notably, one study observed changes in TL when assessing combinations between pain and stress, arguing that a combined effect of the two may accelerate cellular ageing [26]. Chronic pain has been estimated to shorten life by 4–5 years (as measured in loss of base pairs) and when combined with depression, the life-shortening effect adds up to 6 years in total [5]. Hence, research aiming to investigate correlations between emotional well-being, pain experiences and cellular aging are urgently needed and could contribute to new ways of developing and evaluating chronic pain interventions. Health care interventions which include elements of emotional well-being and spiritual health are important in nursing care and are emerging as potential components of health services [27] with presumed positive implications for stress, mental health, pain management [27,28] and even cellular aging.

Spirituality is increasingly studied as a potential factor of resilience in health studies [29,30]. It may reduce the impact of stress and fatigue in cancer patients [31]. Mindfulness, a particular aspect of spirituality, is consistently negatively correlated with depression and anxiety, which again are important elements in the pain experience [32]. In chronically ill patients, spiritual experiences and mindfulness buffer the stress associated with chronic disease [33]. Hence, spirituality seems an important domain to explore further.

Spirituality is a concept, where no accepted definition exists. While definitions of religiosity normally agree on the fact that some concept of a transcendent God and worship is involved [34] spirituality is a broader concept. It ranges from a very broad conceptualization with meaning and purpose at its core to a narrower definition, where some aspect of a sacred and transcendent reality and a personal relationship is mentioned [30]. While most research in this area stems from the USA, where religious involvement is much stronger, more recent approaches, especially those from Europe, focus more on a generic kind of spirituality. In a more secular country like Sweden, it is more likely that the concept of spirituality will tap into the resources that interest us. Of note, there has been two interesting predecessor studies [35,36]. These studies showed that telomere length is associated with religiosity and religious involvement. Especially the study by Hill et al [36] is relevant for our context. It showed that in a probability sample religiosity is a robust predictor of telomere length. Models of how spirituality/religiosity might impact health contain, among others, the idea that there might be a direct causation. As religiosity/spirituality impacts allostatic load, stress and stress-behavior, and in consequence also inflammatory markers [37], one potential marker of this effect might be telomere length. As chronic pain constitutes an important contributor to psychological stress and allostatic load, we reasoned that telomere length and telomerase activity might be good proxy measures for cellular ageing.

Thus, the aim of this study was to explore correlations of Patient Reported Outcome Measurements [PROM's] and markers of cellular ageing among patients with chronic non-malignant pain. We used PROM's of clinical pain, physical and emotional functioning, spiritual, religious and personal beliefs, mindfulness and quality of life. We used TL and TA as measures of cellular ageing and potential mediators between PROMS and cellular ageing were explored.

## Methods

### Participant recruitment

The participants in this study were individuals suffering from chronic pain conditions referred to tertiary specialist centers and enrolled at either the Vidar Clinic, Järna *or* the Pain and Rehabilitation Center, University Hospital, Linköping between June 2015 and December 2017. The study was preregistered (ClinicalTrials.gov Identifier NCT02459639), and ethically approved by the Regional Ethics Committee in Stockholm (EPN 2014/953-31/1). After oral and written information, participants made an informed decision about whether to participate or not with

no implications for their care. Inclusion criteria were ICD-10 M79, *OR* chronic pain (>3 months) in neck/shoulders and/or lower back or generalized pain (e.g. fibromyalgia), ≥18 years of age, fluent in Swedish, resident in Stockholm, Sörmland or Östergötland county councils. Comorbidities were allowed apart from psychotic illness, addiction, and cancer.

Altogether, 57 patients at the Vidar Clinic and 138 patients at Linköping were invited to participate in the study, of which 51 and 35 patients respectively showed interest to participate. Of these, 51 and 33 patients respectively met the inclusion criteria and agreed to participate. The final analysed sample consisted of 84 patients. The Vidar Clinic had a smaller number of patients under treatment than the Rehabilitation Center, University hospital, Linköping had, which explains the differences in numbers of recruited participants.

## Data collection

This was an explorative pilot study with cross-sectional design, assessing clinical characteristics and correlations in a sample of persons with chronic non-malignant pain. Outcome measures were collected after admission. Because of its exploratory nature, no sample size was calculated but a minimum of 70 enrolled patients was strived for, which allows to document an effect size of r = .3 with 72% power and reaches a power of 80% with r = .32. These are the ranges of correlations that were deemed likely and clinically important. Patients were recruited from two geographic locations–one with an out-patient setting (Linköping) and one with an in-patient setting (the Vidar Clinic). The same standard protocols for blood sampling were used at both sites, with the patients in a fasting state and where samples were collected after admission. Questionnaires were filled out at the time of the blood sampling. Data was entered twice for quality assurance and the completed database was then validated by cross-checking 12% of all data, whereby 0.7% were found to be incorrect due to small systematic errors. Those errors were controlled and adjusted for in the whole dataset before analysis.

## Measurements

The outcome measures were TA, TL and PROM's of clinical pain, physical and emotional functioning, spiritual, religious and personal beliefs, mindfulness and quality of life. In addition, demographics, anthropometrics, blood pressure and blood samples were collected. A BMI between 18.5–24.9 kg/m$^2$ would be considered normal, as was a blood pressure <140/90 mmHg. The participants' blood samples were sent for standard clinical analysis at the Clinical Chemistry Laboratory at Linköping University Hospital and at Karolinska University Laboratory at Södertälje Hospital. The blood count was measured as well as the properties of erythrocytes, leucocytes, thrombocytes, standard lipid status (triglycerides and cholesterol with quotas) and inflammatory profile (sedimentation rate, C-reactive protein). Also, levels of haemoglobin, glucose, sodium, potassium and creatinine were assessed. The IMMPACT recommendations for core outcome measurements in chronic pain clinical trials [38] were considered and applied heading-wise, but not strictly followed in regard to the scales used. Some questionnaries were replaced due to having been validated in Sweden and others were added due to relevance for this study.

*Clinical pain* was assessed by means of intensity using the Numerical Rating Scale [NRS] (0 = no pain, 10 = worst possible pain) which is a commonly used instrument, included in the Swedish National Quality Registry for Pain Rehabilitation [SQRP] and recommended by IMMPACT [29]. A score <4 translates to mild pain, 4–6 moderate and >7 severe pain. Pain duration (years, months and days) and pain spread was also assessed by means of well established self-report questionaire from SQRP. In addition, the Chronic Pain Acceptance Questionnaire [CPAQ] (α ≥ .80), with the subscales of *pain willingness* and *activity engagement* was

used because of its emphasis on aspects of coping, which is a focus area within pain research [39,40]. In accordance with recommendations from Dworkin et al [38], patient use of rescue analgesics (fast-acting medications) was registered. *Physical functioning* was evaluated with the Swedish National Board of Health and Welfare's indicator questions on physical activity [FYS], recently validated [41], as well as EUROQOL 5 Dimensions-3L [EQ5D] ($\alpha > .85$) [38]. The latter is an established measure of quality of life, including aspects of physical functioning. The EQ5D index value is a single number calculated from the different health states obtained within the questionnaire. This number is commonly ranging from 0 to 1, although negative numbers are possible, and it reflects how good or bad the health state is in comparison with the general population in a specific country. The EQ5D VAS is a single question with the score range 0–100, ranging from worst possible to best possible health [42]. *Emotional functioning* was measured with the Hospital Anxiety and Depression Scale [HADS] ($\alpha > .89$), a reliable and valid instrument with subscales measuring *depression* and *anxiety* [43], validated in Sweden [44] and where high values indicate psychological distress [45]. *Participant ratings of global improvement and satisfaction with treatment*, as well as *Symptoms and adverse events* were assessed qualitatively, and will be reported separately.

Spiritual, religious and personal beliefs, are increasingly recognized as a distinctive, important, and cross-cultural concepts in quality of life assessments [46]. Hence, in addition to the IMMPACT recommendations we assessed spiritual, religious and personal beliefs using the WHO Quality of Life, Spiritual, Religious and Personal Beliefs (WHOQOL-SRPB) ($\alpha = .85$) questionnaire in its appropriate Swedish adaptation [47]. The questionnaire has the following eight subscales: *spiritual connection*, *meaning and purpose in life*, *experiences of awe and wonder*, *wholeness and integration*, *spiritual strength*, *inner peace*, *hope and optimism*, *and faith*. Mindfulness was assessed using the Five Facets Mindfulness Questionnaire [FFMQ]) ($\alpha = .81$) in order to complement the WHOQOL-SRPB to assess whether improvements in health could be mediated by increased mindfulness. The FFMQ has five subscales: *non-reactivity*, *observing*, *acting with awareness*, *describing* and *non-judging*. The FFMQ is the most frequently used questionnaire for mindfulness [48] and has been validated in Sweden [49]. The information regarding the number of items and min-max score for each questionnaire is found in Table 3.

## Telomerase measurement

Blood samples used for this study were collected between 2015 and 2017 at the two sites. TA was detected by the real-time telomeric repeat amplification protocol (RT-TRAP) [50], with some modifications. In brief, blood samples were drawn in line with Swedish nursing standards, with the participants in a fasting and resting state (minimum 20 minutes). Whole blood was obtained in 8 ml BD Vacutainer® CPT™ Mononuclear Cell Preparation Tube—Sodium Citrate (Becton Dickinson). Thereafter, lymphocytes and monocytes were separated within two hours of sampling, according to the manufacturer's protocol. Approximately half of the pelleted cells were lysed by incubation with 120 µl CHAPS (Merck Millipore, including 0.15 units/µl RiboLock [LifeTechnologies, Thermo Fisher Scientific]) on wet ice for 30 min and with three short vortexes. Following this initial preparation done locally at the laboratories at both respective sites, the samples were labelled and frozen at -80˚C, after which they were transported to Center for Molecular Medicine (CMM) at Karolinska Institutet for storage and further analysis.

Thawed lysate was centrifuged at 4˚C at 12000xg for 20 min and the supernatant was transferred to an empty tube. The total protein concentration was measured in the supernatant by the DC Protein Assay (Bio-Rad) and all patients' cell lysates were diluted to equal concentrations in CHAPS (0.95 µg/µl), aliquoted and refrozen at -80˚C. An aliquot per sample was

thawed and equal amounts of total protein (3.8 μg) from each sample was added to a reaction mix with a total volume of 50 μL containing 1 mM of each dNTP, 20 mM Tris-HCl (pH 8.3), 2.5 mM MgCl2, 63 mM KCl, 0.05% Tween 20, 1 mM EGTA, 8 μmol of each of the primers TS (5'-AATCCGTCGAGCAGAGTT-3') and ACX (5'- GCGCGG (CTTACC) 3CTAACC-3'). HepG2 (hepatic cancer) and ARO (thyroid cancer) cell lines were used as telomerase-positive controls, whereas CHAPS buffer and heat-inactivated samples were used as negative controls. TSR8 is an oligonucleotide with a sequence identical to the TS primer extended with 8 telomeric repeats being AG(GGTTAG)$_7$. Serial dilutions of TSR8 control template were used to generate a standard curve to calculate TA. The serial dilutions were 0.2 amoles/μL, 0.04 amoles/μL, 0.008 amoles/μL, 0.0016 amoles/μL, 0.00032 amoles/μL, and 0.000064 amoles/μL corresponding to 200, 40, 8, 1.6, 0.32 and 0.064 TPG units/μL; TPG is the total product generated, corresponding to the number of TS primers (1 unit equals $10^{-3}$ amoles or 600 molecules) that are extended with at least 3 TTAGGG repeats by telomerase in the extract in a 30 min incubation at 30˚C. The reaction mix was incubated at 30˚C for 30 min followed by termination at 95˚C for 5 min. Then 8 μL of the telomeric repeat products (i.e. corresponding to 8/50*3.8 = 0.608 ug total protein) were used for the RT-TRAP assay amplified by 8 μL Power SYBR Green in 384-well plates. The reaction was performed on QuantStudio 7 Flex (Applied Biosystems; Life Technologies; Thermo Fisher Scientific Inc.) with the following conditions: 95˚C for 10 min, followed by 36 repeats of 95˚C for 20 s, 52˚C for 30 s and 72˚C for 60 s. Samples, controls and standard curve dilutions were run in triplicate, standard curve and controls on all plates. All samples from each patient were run on the same randomized plate. Efficiency was 99–105%. The mean of the correlation coefficients of the standard curves was > 0.993. The mean coefficients of variation (CV) of intra-plate Ct values for the standard dilutions of the five plates was CV = 0.56%, and the mean inter-plate CV of three inter-plate controls was CV = 1.6%. The detection success rate was 100%. For each supernatant, aliquots are still there with original protein concentration.

## Telomere length measurement

The other half of the pelleted cells, collected and prepared at both sites, were also stored at -80˚C until DNA extraction. Genomic DNA was extracted according to the kit instructions [51], but the speed of centrifuge was changed to 6000g to avoid vortex by inverse, and incubated at 37˚C for 3 hours (DNeasy® Blood & Tissue Kit (Qiagene)). DNA concentration was quantified with a NanoDrop ND-1000 Spectrophotometer (Nano-Drop Technologies Inc., Wilmington, DE, USA). Relative TL was determined using real-time quantitative polymerase chain reaction (qPCR) according to Cawthon et al's protocol [52] where the relative telomere to single copy gene (T/S) ratio was determined using a standard curve. In brief, each DNA sample (10 ng) was assessed for the telomere (*Tel*) and the single-copy gene (hemoglobin-b, *HGB*) in triplicate within the same 384-well plate, amplified by using Power SYBR Green in 10 μl total reaction volume. The reaction was performed on QuantStudio 7 Flex (Applied Biosystems; Life Technologies, Carlsbad, CA, USA) with the following conditions: 50˚C for 2 min, then 95˚C for 10 min, followed by 40 repeats of 95˚C for 15 s and 60˚C for 1 min, followed by a dissociation stage to monitor amplification specificity. The same standard curve of pooled DNA from these patient samples ranging from 80 ng to 0.128 ng, was run on each plate for both genes and was used to determine the quantity of each gene for each sample. This allowed controlling for differences in the efficiencies between that of *Tel* and *HGB*. The gene quantities were then used to determine the T/S ratio for each sample. DNA samples with a Ct standard deviation of $\geq 0.35$ between triplicates or a Ct value outside the standard curve were omitted from the analyses. The correlation coefficients of the standard curves were above 0.99 for each

primer set and 384-plate. The inter-plate coefficient of variation (CV) of T/S ratio was 9.1% calculated from a patient sample run in seven 384-well plates. The TL analysis detection success rate was 100%. The primer sequences were (written 5'à3'): Tel1: `CGGTTTGTTTGGGTTTG` `GGTTTGGGTTTGGGTTTGGGTT`; Tel2: `GGCTTGCCTTACCCTTACCCTTACCCTTACCCTTAC` `CCT`; *HGB* Fw: `GCTTCTGACACAACTGTGTTCACTAGC`; *HGB* Rv: `CACCAACTTCATCCACGT` `TCACC`. Samples from all three timepoints per person were assayed in the same 384-well plate.

## Statistical analysis

The software used for analysis was IBM SPSS version 26. Background data is described as means and standard deviations, or frequencies and percentages, and Chi-Square or Fishers exact tests were performed for descriptive background measures. The TA variable was ln-transformed due to significant skewness. Partial correlations were used to assess how TL (adjusted for age) and TA correlated with pain intensity, spread and psychological outcomes at baseline. In addition, we used an analytic approach similar to Hassett et al [5], where the participants were divided into "high pain/low pain"; "high HADS score/low HADS score"; and "high WHOQOL-SRPB score/low WHOQOL-SRPB score" (the group mean score was used as cut-off), and ANCOVA analyses adjusted for age were conducted with TL and TA as dependent variables. A potential mediation model was conducted in an exploratory fashion following the standard mediation model of Baron and Kenny [53]. Direct effects were modeled in univariate regression models. Beta-weights, i.e. partial correlations coefficients were given for the different paths. This model was an ad-hoc decision after we saw the data pattern, because we wanted to explore the data further. It should therefore be viewed as a preliminary additional analysis solely aiming to clarify relationships. We mark significance levels according to a p-level of .05. Corrections for multiple testing were not considered, as this was an exploratory study and no confirmatory statistics had been stipulated in the protocol. Due to the exploratory nature of the study, we do not emphasize significances, but rather effect size, i.e. the extent of the correlation [54].

## Results

### Sample characteristics

The socio-demographic background variables and pain variables of participants are presented in Table 1. The majority of the participants were female, from Sweden and currently working. Most had upper secondary school or university education. Regarding pain, 82% reported it as constant and high, with an average pain intensity of 7.1 (out of 10) and a mean duration of nearly 12 years. Among the participants, 60% were taking prescription drugs for pain and various other types of medications. Almost one third of the participants were using herbals and other supplements. In order to assess representativeness of our sample it was compared to the target population. Table 2 presents essential characteristics of the sample and target population, with target population numbers obtained from a report by SQRP [55].

### Patient reported outcome measures

All PROM's were normally distributed. The PROM's for *clinical pain* including pain intensity, pain duration, pain spread, and medication are presented in Table 1. All other PROM's, including subscales are presented in Table 3. To better understand the interrelatedness of different PROM's, correlations were calculated amongst all the questionnaires and pain measures and are presented in Table 4 and elaborated on in the discussion section.

**Table 1. Socio-demographic background and pain variables of participants.**

| Variable | n | Score | SD |
|---|---|---|---|
| *Gender* | 84 | | |
| **Female** | 84 | 85.7% | |
| **Male** | 84 | 14.3% | |
| *Age* | 84 | 47.6 | 10.3 |
| *Country of birth* | 84 | | |
| **Sweden** | 84 | 77.4% | |
| **Other Nordic countries** | 84 | 3.6% | |
| **Europe** | 84 | 9.5% | |
| **Outside Europe** | 84 | 9.5% | |
| *Education* | 84 | | |
| **Primary schooling** | 84 | 7.1% | |
| **Upper secondary school** | 84 | 64.3% | |
| **University** | 84 | 22.6% | |
| **Other** | 84 | 6.0% | |
| *Working* | 82 | 87.8% | |
| *Blood pressure* | 77 | | |
| **Systolic** | 77 | 120.4 mmHg | 17.3 |
| **Diastolic** | 77 | 76.9 mmHg | 10.8 |
| *BMI* | 77 | 27.6 | 10.3 |
| *ICD-10 diagnosis* | 84 | | |
| **M-diagnoses** | 84 | 66.7% | |
| **R-diagnoses** | 84 | 19.0% | |
| **Q-diagnoses** | 84 | 8.0% | |
| **F-diagnoses** | 84 | 7.1% | |
| **Other diagnoses** | 84 | 2.4% | |
| *Medication* | 84 | | |
| **Prescription drugs for pain** | 84 | 61% | |
| **Prescription drugs for other** | 84 | 45.2% | |
| **Non-prescription drugs for pain** | 75 | 28% | |
| **Non-prescription drugs for other** | 75 | 10.7% | |
| **Herbals** | 77 | 26% | |
| *Type of pain* | 83 | | |
| **Constant** | 83 | 81.9% | |
| **Periodic** | 83 | 18.1% | |
| *Pain duration* | 81 | 11.8 | 10.5 |
| *Pain intensity NRS* | 83 | 7.1 | 1.5 |
| *Pain spread* | 83 | 19.1 | 9.0 |

BMI: Body Mass Index, ICD-10: International Classification of Diseases, M-diagnoses: Diseases of the musculoskeletal system and connective tissue, R-diagnoses: Symptoms, signs and abnormal clinical and laboratory findings not elsewhere classified, Q-diagnoses: Congenital malformations, deformations and chromosomal abnormalities, F-diagnoses: Mental, Behavioral and Neurodevelopmental disorders, NRS: Numerical rating scale.

## Cellular measurements

The mean TL of the group was 1.163 (n = 84, SD = 0.412) and the mean TA (ln transformed) was 2.064 TPG units/μL (n = 80, SD = 1.774). At group level, all routine blood test results were within normal reference intervals according to Swedish standards at Karolinska University Laboratory.

**Table 2. Essential sample characteristics in comparison with the target population.**

| Variable | Sample score | Target population score* |
|---|---|---|
| **Age** | 47 | 44 |
| **Gender F/M** | 86/14% | 79/21% |
| **Working now** | 88% | 67% |
| **Nationality Sweden/other** | 81/19% | 82/18% |
| **Senior high school** | 64% | 41% |
| **University** | 23% | 42% |
| **BMI** | 28 | 27 |
| **Pain duration** | 12 years | 9 years |
| **Pain intensity** | 7.1 | 4.1 |
| **Pain spread** | 19 | 15 |

*Based on Swedish National Quality Registry for Pain Rehabilitation 2020.

## Telomere measurement correlations and associations with PROMs

TL was negatively associated with age (r = -.210, p = .055) and in an age-adjusted correlation, only P-cholesterol (r = .225, p = .041) and LDL-cholesterol (r = .218, p = .050) correlated with TL. No correlations were found between measures in blood samples and TA. In Table 5,

**Table 3. Patient reported outcome measures including subscales and scoring.**

| Questionnaire | n | Items, n | Scoring | Min-max | Mean | Std. Dev |
|---|---|---|---|---|---|---|
| **HADS score** | 84 | 14 | 0–3 | 0–42 | 19.17 | 8.49 |
| HADS anxiety | 84 | 7 | 0–3 | 0–21 | 10.13 | 5.10 |
| HADS depression | 84 | 7 | 0–3 | 0–21 | 9.04 | 4.33 |
| **EQ5D VAS** | 82 | 1 | 0–100 | 0–100 | 40.77 | 17.37 |
| **EQ5D Index value** | 84 | 5 | 1–3 | 5–15 | 0.26 | 0.32 |
| **FFMQ score** | 80 | 29 | 1–5 | 29–145 | 91.95 | 15.12 |
| FFMQ non-reactivity | 80 | 6 | 1–5 | 6–30 | 17.80 | 3.94 |
| FFMQ observe | 80 | 7 | 1–5 | 7–35 | 24.40 | 5.02 |
| FFMQ acting with awareness | 80 | 5 | 1–5 | 5–25 | 14.55 | 3.87 |
| FFMQ describe | 80 | 6 | 1–5 | 6–30 | 19.26 | 4.80 |
| FFMQ non-judging | 80 | 5 | 1–5 | 5–25 | 16.03 | 4.22 |
| **CPAQ score** | 79 | 8 | 0–6 | 0–48 | 22.14 | 7.18 |
| CPAQ pain willingness | 79 | 4 | 0–6 | 0–24 | 11.72 | 5.34 |
| CPAQ activity engagement | 79 | 4 | 0–6 | 0–24 | 10.42 | 4.63 |
| **WHOQOL-SRPB score** | 75 | 32 | 1–5 | 32–160 | 92.00 | 26.44 |
| WHOQOL-SRPB spiritual connection | 75 | 4 | 1–5 | 4–20 | 10.27 | 4.98 |
| WHOQOL-SRPB meaning and purpose in life | 75 | 4 | 1–5 | 4–20 | 12.85 | 3.49 |
| WHOQOL-SRPB experiences of awe and wonder | 75 | 4 | 1–5 | 4–20 | 10.52 | 4.62 |
| WHOQOL-SRPB wholeness and integration | 75 | 4 | 1–5 | 4–20 | 12.84 | 3.83 |
| WHOQOL-SRPB spiritual strength | 75 | 4 | 1–5 | 4–20 | 12.92 | 3.23 |
| WHOQOL-SRPB inner peace | 75 | 4 | 1–5 | 4–20 | 10.93 | 4.04 |
| WHOQOL-SRPB hope and optimism | 75 | 4 | 1–5 | 4–20 | 10.51 | 3.38 |
| WHOQOL-SRPB faith | 75 | 4 | 1–5 | 4–20 | 11.16 | 3.37 |

HADS: Hospital anxiety depression Scale, EQ5D (VAS): Euro Quality of Life 5 Dimensions (visual analogue scale), FFMQ: Five Facets Mindfulness Questionnaire, CPAQ: Chronic Pain Acceptance Questionnaire, WHOQOL-SRPB: WHO Quality of Life–Spiritual, Religious and Personal Beliefs.

**Table 4. Correlations amongst questionnaires and pain measures.**

| | HADS score | EQ5D index | EQ5D VAS | FFMQ score | CPAQ score | WHO score | Pain intensity | Pain duration | Pain spread |
|---|---|---|---|---|---|---|---|---|---|
| HADS score | | | | | | | | | |
| EQ5D index | -.300* | | | | | | | | |
| EQ5D VAS | -.347* | .378* | | | | | | | |
| FFMQ score | -.470* | .019 | .019 | | | | | | |
| CPAQ score | -.292* | .212 | .366* | -.005 | | | | | |
| WHO score | -.347* | .189 | .236* | .485* | .064 | | | | |
| Pain intensity | .292* | -.333* | -.270* | -.230* | -.160 | -.235* | | | |
| Pain duration | .011 | .051 | .004 | .070 | -.101 | .166 | -.108 | | |
| Pain spread | .298* | -.104 | -.158 | .008 | -.273* | .090 | .095 | .330* | |

**Abbreviations:** Hospital Anxiety and Depression Scale (HADS), Euroqol 5 Dimensions (EQ5D), Visual Analogue Scale (VAS), Five Facets Mindfulness Questionnaire (FFMQ), Chronic Pain Acceptance Questionnaire (CPAQ), WHO Quality of Life Spiritual, Religious and Personal Beliefs (WHO).

*Correlation is significant at the 0.05 level (2-tailed).

correlations with PROM's are presented, where a negative correlation was found between the TL and HADS score (r = -.219, p = .047). Also, a positive correlation was found between the TL and WHOQOL-SRPB (r = .224, p = .052) (Table 5). Interestingly, TL did not correlate with any of the pain measures. TA did not correlate with any of the self-reported questionnaires but with pain spread (r = .222, p = .049). Further, the groups were divided into "high pain/low pain", "high HADS score/low HADS score" and "highWHOQOL-SRPB/lowWHO-QOL-SRPB" (the group mean score was used as cut-off), and ANCOVA analyses adjusted for age were conducted with TL and TA as dependent variables. This was in order to assess whether dichotomised variables of pain, anxiety/depression, and SRPB would yield additional information about the associations with TL and TA, to what can be found through correlations with continuous variables. However, no associations were detected in this additional analysis (for TA and HADS: $Eta^2 = 0.00$; p = 0.901; for TL and HADS: $Eta^2 = 0.012$; p = 0.32; for TA and pain: $Eta^2 = 0.011$, p = 0.357; for TL and pain: $Eta^2 = 0.001$; p = 0.78; for TA and WHO-QOL-SRPB: $Eta^2 = 0,008$; p = 0,460; for TL and WHOQOL-SRPB: $Eta^2 = 0,022$; p = 0,225).

## Mediation effects

Since the ANCOVA-analysis with dichotomized SRPB scores might have collapsed variance, and as the first order correlations and theoretical consideration suggested that the relationship between spiritual well-being (WHOQOL-SRPB) and TL might be mediated by the effect of the

**Table 5. Partial correlations with p-values, between TA/TL and self-report measures.**

| | HADS | EQ5D index | EQ5D VAS | FFMQ | CPAQ | WHO | PI | PD | PS |
|---|---|---|---|---|---|---|---|---|---|
| TL | -.219 (.047)* | -.003 (.977) | .141 (.211) | .131 (.250) | .189 (.089) | .224 (.052) | -.154 (.166) | .083 (.462) | -.188 (.090) |
| TA | -.042 (.709) | -.080 (.480) | -.100 (.383) | .039 (.738) | -.031 (.787) | .145 (.225) | -.177 (.119) | .181 (.113) | .222 (.049)* |

**Abbreviations**: Hospital Anxiety and Depression Scale (HADS), Euroqol 5 Dimensions (EQ5D), Visual Analogue Scale (VAS), Five Facets Mindfulness Questionnaire (FFMQ), Chronic Pain Acceptance Questionnaire (CPAQ), WHO Quality of Life Spiritual, Religious and Personal Beliefs (WHO), Pain Intensity (PI), Pain Duration (PD), Pain Spread (PS), Telomere Length (TL), Telomerase Activity (TA).

*Correlation is significant at the 0.05 level.

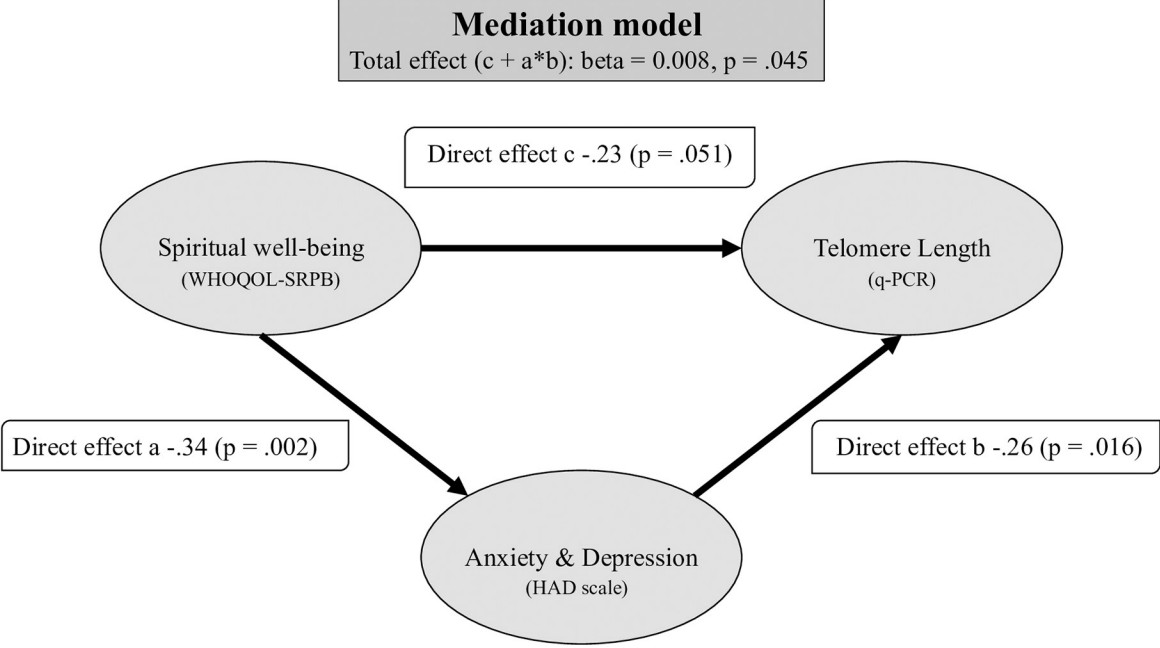

WHOQOL-SRPB: WHO Quality of Life –Spiritual, Religious and Personal Beliefs, HAD scale: Hospital Anxiety and Depression Scale; q-PCR: quantitative Polymerase Chain Reaction.

**Fig 1. Mediation model on spirituality, depression and anxiety, and telomere length.**

former on depression and anxiety (HADS), we tested a potential mediation model in an exploratory fashion. First, the direct effects a, b and c were modeled in univariate regression models. Beta-weights, i.e. partial correlations coefficients are given for the different paths in Fig 1. Modelling the total effect using the mediation path, the interaction a*b (Total Effect = c + a*b), following the standard mediation model of Baron and Kenny [53] yields a total effect (beta = 0.008; p = .045), i.e. supports the idea of a mediation of the direct effect of spiritual well-being on TL by anxiety and depression (HADS). However, the indirect effect, the interaction a*b (ab = 0.002; beta = -.23; p = .15) is not significant.

## Discussion

To the best of our knowledge this is the first study that explores the body-mind-spirit perspective of pain in an integrated and comprehensive manner, by investigating telomeres, blood tests and questionnaires covering pain, coping, depression, mindfulness and spiritual, personal and religious beliefs.

Table 3 presents the results from all PROM's and can be interpreted as follows. A typical participant in this study was a woman who rated her overall health poorly, with an EQ5D VAS mean score of 41 out of 100 maximum. In comparison, the non-standardized Swedish population norm is 83 out of 100 [56]. The EQ5D index value is 0.85 for the Swedish general population, whereas our participant mean score was 0.26 (SD 0.32). An average participant in this study had a clinically probable risk of both anxiety and depression with a moderate pain coping capacity–mean scores were higher for pain willingness than for activity engagement. When facets of mindfulness were reported, our average participant was struggling with acting with awareness or staying focused on the task at hand without becoming distracted. Mindfulness meditation could be beneficial as an intervention for this population since it can improve

pain, depression symptoms and quality of life with no risk of addiction [57]. On average, participants scored moderately on all facets of the WHOQOL-SRPB. Their sense of meaning and purpose in life together with a feeling of wholeness and integration was slightly more prominent while there was a lack of hope and optimism as well as few experiences of awe and wonder. Despite the obvious challenges to develop a universal instrument for spiritual, religious and personal beliefs that allows comparison among people with different worldviews [58] the SRPB dimensions used in our study can be seen as related to spiritual quality of life in one way or another. Studies have repeatedly established positive links between spirituality, religion and health. As an example, religious involvement is correlated with better mental health in the areas of depression, substance abuse, and suicide [59]. It has also been shown that cardiovascular and cortisol stress responses of a sample of university students can be predicted by secularity, religiosity, spirituality, and existential search [60]. Religiosity was an independent predictor of telomere length in a probability sample [36]. Since spiritual, religious and personal beliefs seem to enhance quality of life and affective well-being in chronic pain patients, partly mediated through coping [61–63], health care interventions including dimensions of spiritual, religious and personal meaning might be beneficial.

As stated in the results, interestingly, the blood samples did not deviate from normal reference intervals indicating that patients with severe and long-lasting pain paradoxically appear "perfectly healthy". However, more complex proteomics or pro-inflammatory markers have been shown to deviate among chronic pain patients and correlate with other health measures such as HADS scores [64].

The unbalanced gender ratio (See Table 1) in this sample is noteworthy, and characteristic of the pain patient population nationally as well as globally, illuminating a gender issue. Women are notably overrepresented among patients living with chronic pain [65–67] and pain has been highlighted as a gendered condition, possibly subject to medical gender bias [66]. Further adding to this, a report from 2018 requested by the Swedish Government concluded that there were close to zero (1 of 185) reviews with focus on women on the topic of this female dominated condition [67].

As presented in Table 4, the participants' self-reported anxiety and depression had stronger correlations with measures of mindfulness and spiritual/religious/personal well-being than with pain measures. Interestingly, the single question EQ5D VAS had nearly as many correlations of moderate strength as the HADS. That is, when participants in this study scored their overall health with a single number it correlated with almost all other self-report measures. Indeed, our findings indicate that stronger spiritual, religious or personal beliefs may be protective of depression and anxiety, and that this may perhaps be mediated by increased mindfulness—presence in everyday life. However, our study design does not allow us to assess the direction of association, whereby it may also be so that decreased depression and anxiety instead enables increased mindfulness and spiritual, religious and personal beliefs.

One interesting finding in this study is that the WHOQOL-SRPB had a positive correlation to TL (See Table 5), meaning that a greater sense of spiritual well-being, meaning, purpose and wholeness might be linked with longer telomeres. Furthermore, our exploratory model (Fig 1) supports the idea of mediation of the direct effect of spiritual well-being on TL by anxiety and depression as measured by the HADS. The relationship between spiritual well-being and TL could indicate a weak relationship between PROM's and aspects of telomeres. As this study consisted only of a small sample, some correlations, though interesting from a conceptual point of view, did not reach conventional significance thresholds. This likely indicates a power problem which underlines the need for replication of the study with a larger data set. As seen in Table 5, the HADS was negatively correlated with the TL, meaning that higher depression/anxiety scores, could be linked to shorter telomeres. Similar findings have been described by

Hassett et al [5] and Sibille et al [26]. One interesting finding of our study, supported by independent evidence [68], is that spirituality might be an important factor in the maintenance of health via longer telomeres to be further explored in larger and more focused studies. Our study had a broad focus on various PROMs. It might be worthwhile to study the question more specifically whether spirituality as a part of religiosity is associated with longer telomeres, as this might be one of the biological mechanisms how spirituality/religiosity causes better health outcomes that have been widely described [37].

Another finding presented in Table 5 is the absence of correlation between TL and the pain measures in this study. This is surprising since others have found statistically significant correlations between TL and pain measures [5,21–24]. In contrast, TA had a positive correlation with pain spread, i.e. high TA was associated with higher pain spread. This is ambiguous since TL is negatively associated with pain spread to a notable degree. Surprisingly, when repeating the cross table statistical analysis of Hassett et al [5] by grouping high pain-high depression in contrast to low pain-low depression we were still unable to reproduce the results with correlations between TL and high depression—high pain. For all lab tests, pain measures and other self-report measures it is the HADS and the WHOQOL-SRPB that emerge in correlation with TL (Table 5).

In summary, what ultimately correlates with the genomic integrity within this sample of chronic pain patients is neither quantitative pain measures or scales, nor index numbers generated for health economy calculation purposes, but instead anxiety, depression and spiritual, religious and personal beliefs. And those, in turn, have a significant relation to both the single EQ5D VAS question and some of the self-report pain measures (see Table 4). The way pain patients feel is often dismissed in health care [8,9]. Our study suggests that the way how patients feel is actually reflected in the state of genomic integrity. It makes also understandable a well-known phenomenon, namely that the single best predictor of longevity is how patients describe their status of health [10,69].

Although the recommendations by large organizations like IMMPACT are to use NRS as the core measurement for clinical pain [38], it doesn't relate to the genomic integrity of the participants in this study. And while measures of emotional functioning are included in the recommendations for core outcome measures, the spiritual/religious/personal beliefs dimension of pain is absent from the international recommendations [38].

The global opioid crisis is a painful reality that is of foremost relevance to pain management as well as for this study. Opioids are commonly prescribed to chronic pain patients in spite of doubtful effectiveness as long-term treatment. Between 1999–2014 approximately 165 000 persons died of overdose related to opioid medications—only in the United States [70]. Moreover, during 2015 opioids were responsible for a third of the almost half a million global drug deaths [71]. Despite the relation to global morbidity, mortality and doubtful long-term effectiveness, 24% of the participants in our sample of chronic pain patients had opioids prescribed for their pain. However, our study gives reasons to suggest that it might be beneficial for health care to be guided by interventions also including the dimensions of emotional well-being and spiritual, religious and personal beliefs when it comes to pain management, i.e. a focus on promoting connection instead of addiction.

Increasingly, more patients are offered rehabilitation in so-called multimodal programs with psychological and physical activities, and this biopsychosocial approach and model in pain management is currently recommended in Sweden [64,72], though without reference to spiritual/religious/personal beliefs aspects of health and pain. Given that TL is a good indicator for longevity and robustness of health [12] and that spiritual/religious/personal well-being is associated with TL this indicates a novel arena with potential in pain rehabilitation. Our findings are preliminary and warrant further scrutiny.

## Strengths and limitations

This study has a number of strengths as well as limitations. The double data entry and cross-checking and careful validation of the database before analysis improved methodological strength in this study. Also, the inclusion of two geographically separate locations as study sites with the aim of improving generalizability can be seen as a strength. In addition, since Complementary and Alternative Medicine (CAM) modalities are widely used by pain patients [73], the inclusion of The Vidar Clinic as a formal health care provider including certain CAM therapies can also been seen as a strength contributing to generalizability. Moreover, this study assessed the levels of biomarkers that are known predictors for several age-related diseases and mortality and of cumulative life stress and age-related diseases. The use of objective biomarkers in addition to PROMs can be considered a strength of the study. Furthermore, the initial preparation of the blood samples used for TL and TA measurement was performed by different staff at the two sites, and a standard protocol was followed in order to increase reliability of the measurements across sites. Therefore, when conducting analyses of the mononuclear cells in laboratory environment, the quality of blood sampling and initial preparation could be validated. Additionally, the randomisation of the telomere samples as well as blinding of laboratory personnel performing the tests added methodological strength and reduced bias.

A main limitation with this study is its cross-sectional design with the innate inability to draw causal conclusions, and its small sample size. The latter was due to organisational restrictions that could not be remedied. However, the goal was rather to generate some preliminary insights, questions and hypotheses for future studies.

There are several possible sources of bias in this study. First, the two sites had different participation rates which can generate bias. Altogether, 45% of all eligible persons accepted participation, which incurs a risk of selection bias and raises the concern of the sample not being representative of the target population. Yet, by comparing our data to national health registry data [54] (see Table 2) we can see that our sample was in fact representative of the target population based on the demographic data that was collected.

Another limitation is that other potential confounders apart from BMI, age or sex may have accounted for some of the results. We did not investigate the effect of tobacco or alcohol use, which are known to confound telomere measurements. In addition, levels of exercise and nutrition and patients' medication should also be considered as possible confounders.

The magnitudes of correlations that were found were rather small, between r = .2 and .3. Due to the small sample size and the comparatively large number of correlations, inherent to the exploratory nature of the current project, no final conclusions can be drawn from the findings.

## Implications for further studies

Therefore, it would seem natural to improve on this study with a larger sample in order to assess whether the positive association between TL and spiritual well-being can be sustained, as well as the negative association between depression and anxiety and TL. Furthermore, the positive correlation between TA and pain spread warrants further scrutiny. The former would be indicative of a protective effect of spirituality and meaning-making against physical disintegration. The latter would point to an association between repair processes and the spreading of pain. With such focused hypotheses further studies could produce more confirmatory evidence than our data can provide.

Also, it should be noted that the mediation model is intuitive, but far from definitive. The suggestion from our findings that SRPB together with an interaction between SRPB and HADS should have a positive, albeit small, influence on telomere length certainly warrants

further scrutiny. Further studies should include longitudinal data lending itself to a clear causal analysis, which our cross-sectional data cannot provide. Furthermore, one can argue that our study results demonstrate mediation only according to the original Baron and Kenny approach, but not according to more recent discussions [74,75], which would require an independent interaction term to be significant. However, we think that this potential mediation might be interesting for other researchers to scrutinize in more detail.

## Conclusion

In summary, this study conducted with chronic pain patients confirms the view of a complex and holistic web of health dimensions that are affected by and associated with their suffering, as well as a heavy symptom burden. Our findings show that telomeres tend to be shorter in more anxious and depressed pain patients, but not in patients with more self-reported pain. Intriguingly, spiritual/religious/personal well-being as measured with WHOQOL-SRPB, was in this sample of patients correlated with TL. TA was more pronounced in patients reporting higher pain spread. These findings are preliminary and need confirmation. This study confirms the notion that numerical estimations of pain may be too crude to capture the complexity of chronic pain, and the widespread use of such measures in health care and research can possibly be explained by a historical tendency to assess the effectiveness of pharmacological pain management. Considering the international opioid crisis, non-pharmacological and non-opioid treatments emerge as a way forward. Our data support the idea that the emotional and spiritual/religious/personal beliefs dimensions of pain and health are neglected yet highly important aspects for the well-being and longevity of chronic pain patients and hence should be an additional focus in pain management and most importantly for further research.

## Acknowledgments

The many anonymous patients are deeply appreciated for generously giving of body, mind and spirit, making this project possible. Also, the following persons are gratefully acknowledged: Annele Claeson for her general support and efforts during data entry, Eva-Britt Lind and Ulrika Wentzel Olausson for essential contribution in recruitment, data collection, and data entry. Vincent Millisher for substantial contribution in analysing telomere data, as well as editing and reviewing. Tobias Sundberg for contribution to database development and initial crude analysis. Sebastian Sauer for help with evaluating the mediation model. Maria Arman for insights and expertise on nursing and Johanna Hök Nordberg for integrated feedback on manuscript and both for thoughtful supervision. Linköping Smärtcentrum and The Vidar Clinic are acknowledged for smooth collaboration. We are also grateful for initial advice and guidance on genomic aspects of the project by Elizabeth Blackburn. Lastly a thank you to the unnamed reviewers who helped elevate this work.

Torkel Falkenberg passed away before the submission/publication of the final version of this manuscript. Linn Rönne-Petersén accepts responsibility for the integrity and validity for the data collected and analyzed.

## Author Contributions

**Conceptualization:** Catharina Lavebratt, Björn Gerdle, Bijar Ghafouri, Torkel Falkenberg.

**Data curation:** Linn Rönne-Petersén, Harald Walach, Bijar Ghafouri, Torkel Falkenberg.

**Formal analysis:** Linn Rönne-Petersén, Maria Niemi, Harald Walach, Catharina Lavebratt, Liu L. Yang, Björn Gerdle, Bijar Ghafouri, Torkel Falkenberg.

**Funding acquisition:** Björn Gerdle, Torkel Falkenberg.

**Investigation:** Linn Rönne-Petersén, Maria Niemi, Harald Walach, Catharina Lavebratt, Liu L. Yang, Björn Gerdle, Bijar Ghafouri, Torkel Falkenberg.

**Methodology:** Linn Rönne-Petersén, Maria Niemi, Harald Walach, Catharina Lavebratt, Liu L. Yang, Bijar Ghafouri, Torkel Falkenberg.

**Project administration:** Linn Rönne-Petersén, Catharina Lavebratt, Björn Gerdle, Torkel Falkenberg.

**Resources:** Torkel Falkenberg.

**Supervision:** Maria Niemi, Harald Walach, Catharina Lavebratt, Torkel Falkenberg.

**Validation:** Linn Rönne-Petersén, Maria Niemi, Harald Walach, Björn Gerdle, Torkel Falkenberg.

**Visualization:** Linn Rönne-Petersén, Torkel Falkenberg.

**Writing – original draft:** Linn Rönne-Petersén, Maria Niemi, Harald Walach, Catharina Lavebratt, Liu L. Yang, Björn Gerdle, Bijar Ghafouri, Torkel Falkenberg.

**Writing – review & editing:** Linn Rönne-Petersén, Maria Niemi, Harald Walach, Catharina Lavebratt, Björn Gerdle, Bijar Ghafouri, Torkel Falkenberg.

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
