## [Decision Letter · Decision Letter 0]

26 Apr 2023

PONE-D-23-08421Exploring emotional well-being, personal, spiritual and religious beliefs and telomere length in chronic pain patients - A pilot study with cross-sectional designPLOS ONE

Dear Dr. Falkenberg,

Thank you for submitting your manuscript to PLOS ONE. After careful consideration, we feel that it has merit but does not fully meet PLOS ONE’s publication criteria as it currently stands. Therefore, we invite you to submit a revised version of the manuscript that addresses the points raised during the review process.

We look forward to receiving your revised manuscript.

Kind regards,

Alessandro Vittori, M.D.

Academic Editor

PLOS ONE

Journal Requirements:

6. We note you have included a table to which you do not refer in the text of your manuscript. Please ensure that you refer to Table 6 in your text; if accepted, production will need this reference to link the reader to the Table.

Reviewers' comments:

Reviewer's Responses to Questions

**Comments to the Author**

1. Is the manuscript technically sound, and do the data support the conclusions?

Reviewer #1: Partly

Reviewer #2: Partly

2. Has the statistical analysis been performed appropriately and rigorously? 

Reviewer #1: Yes

Reviewer #2: Yes

3. Have the authors made all data underlying the findings in their manuscript fully available?

Reviewer #1: Yes

Reviewer #2: Yes

4. Is the manuscript presented in an intelligible fashion and written in standard English?

Reviewer #1: Yes

Reviewer #2: Yes

5. Review Comments to the Author

Reviewer #1: The study explores a fascinating hypothesis, around the relationship between several psychological states and telomere length. I should note beforehand that my expertise lies in spirituality and psychology, not in biology, so there is a lot about this study that I cannot evaluate. This may also have lead to some misunderstandings on my part, for which I hope the authors will forgive me.

My main question around this study has to do with its rationale. A minor point on this is on page 4, when the authors state that 'sometimes TA is a signal for repair activity, or an indicator of a lack of such activity'. This immediately raised the question for me what then the relevance and validity of this measure is. Why is it included in this study? When is or isn't it an indicator for (non-)activity? Perhaps the authors can elaborate on this a little.

Also, from the introduction it doesn't become clear what the rationale behind the study is: Why do the authors expect TL and TA to relate to the mentioned PROs? What's more, from the methods section I understand that this study is part of a larger clinical trial on the effects of anthroposophical integrative care (AIC) or standard care (SC) on TA and TL, with the PROs being secondary outcome variables in this trial. What then, is the reason behind this current study? The research question of this manuscript is also not mentioned in the trial registration. Might it be that the authors were examining bivariate correlations to check their data, stumbled upon the associations reported here and decided to publish about it? If so, the authors should at least make this clear.

Assuming for now that the authors have a clear rationale for examining the research question in this manuscript, I have some further questions for clarification (in order of appearance, not in order of importance):

* What were the arguments for the inclusion and exclusion criteria? Specifically, the county councils and the comorbities (which are also reported much more specifically in the trial registration).

* Why were the different parameters in the blood samples examined? What role do these play in the current study?

* Why were both partial correlations and linear regression analysis examined? Did the first influence which variables were included in the latter? Or was there some other reason?

* Were no control varibales included in the linear regression analysis, particularly gender and level of education? If so, why not? [I do want to complement the authors for controlling for multiple testing]

* How were differences in age, pain duration and pain spread between the samples from the study sites handled?

* What does it mean for the study that neither TL nor TA were correlated to any of the pain measures?

* Why was an ANCOVA performed for the HADS and not the WHOQOL?

* What theoretical considerations underlie the test for the mediation effect?

* In the discussion section the authors discuss many findings that do not pertain to the research question. What is the purpose of this discussion?

* What is the connection of the study with opioid use (p.22-23)? This was not included in the study was it?

Overall, I think the authors present an interesting finding, based on acceptable methodology, that certainly warrants further study. However, the background of the study in insufficiently clear and needs a more sound rationale.

Reviewer #2: Dear authors, thank you for your efforts. Your study covers a relevant topic and an important attempt to promote more holistic non-pharmacological treatments in chronc pain patients. Your methodolocy is sound (scales used, handling of biological materials) but the sample size is very small and relevant confounders have not been considered (tabacco and alcohol as stated by yourself, other lifestyle factors as exercise and nutrition, or patients medication specially anti-inflammatory drugs). This makes the results, specially the associations with TL/TA rather speculative and hypothetical. Associations between spiritual und religious beliefs and mental health (anxiety and depression) are well documented and have been found in several studies and different contexts (pain, cardiac rehabilitation, coping with disease).

6. PLOS authors have the option to publish the peer review history of their article (what does this mean?). If published, this will include your full peer review and any attached files.

Reviewer #1: **Yes: **Anja Visser

Reviewer #2: **Yes: **René Hefti M.D.

---

## [Author Response · Author response to Decision Letter 0]

21 Jul 2023

Response to Reviewers

Torkel Falkenberg

Karolinska Institutet

Dept. of Neurobiology, Care Sciences & Society

Alfred Nobels allé 23, 141 52 Huddinge

Sweden

PONE-D-23-08421

Exploring emotional well-being, personal, spiritual and religious beliefs and telomere length in chronic pain patients - A pilot study with cross-sectional design

May 2023,

Dear Editor, 

Thank you for considering our manuscript for publication in PLOS ONE and please find our revisions and comments that addresses the points raised during the review process as described below.

Reviewer #1: 

*The study explores a fascinating hypothesis, around the relationship between several psychological states and telomere length. I should note beforehand that my expertise lies in spirituality and psychology, not in biology, so there is a lot about this study that I cannot evaluate. This may also have led to some misunderstandings on my part, for which I hope the authors will forgive me.

-Thank you so much for appreciating our work.

*My main question around this study has to do with its rationale. A minor point on this is on page 4, when the authors state that 'sometimes TA is a signal for repair activity, or an indicator of a lack of such activity'. This immediately raised the question for me what then the relevance and validity of this measure is. Why is it included in this study? When is or isn't it an indicator for (non-)activity? Perhaps the authors can elaborate on this a little.

-Thank you for pointing this out. We have tried to clarify this and changed the text to: “Changes in TA enzyme levels have implications for telomere length and protection of the chromosomes [18, 19], making it suitable for examining more immediate effects in time of for example certain lifestyle factors or health-promoting behaviors [20].

*Also, from the introduction it doesn't become clear what the rationale behind the study is: Why do the authors expect TL and TA to relate to the mentioned PROs? 

- We have made the rationale clearer in the introduction. In addition, we point to research that has linked TL to chronic pain and cellular ageing, whereas reports on pain and TA are lacking and hence the inclusion in this study in order to learn more about the possible role of TA levels and pain. 

*What's more, from the methods section I understand that this study is part of a larger clinical trial on the effects of anthroposophical integrative care (AIC) or standard care (SC) on TA and TL, with the PROs being secondary outcome variables in this trial. What then, is the reason behind this current study? 

-Thank you for addressing this. Indeed, this is the baseline reporting (before care programs are provided). In the rationale we describe the need for research aiming to investigate correlations between emotional well-being, pain experiences and cellular aging in the group of chronic pain patients. Provided such correlations exists, this could underpin “new ways of developing and evaluating chronic pain interventions” for this group of patients.

*The research question of this manuscript is also not mentioned in the trial registration. Might it be that the authors were examining bivariate correlations to check their data, stumbled upon the associations reported here and decided to publish about it? If so, the authors should at least make this clear.

-The reviewer is correct. At the time of the trial registration, we did not foresee baseline reporting as a self-standing paper. However, when confronted with the massive amount of data in this project, we realized the need for detailing the pain patients reported PROM’s in relation to TA and TL before they were subjected to care programs, since such information may help the development and evaluation of current pain management.

*Assuming for now that the authors have a clear rationale for examining the research question in this manuscript, I have some further questions for clarification (in order of appearance, not in order of importance):

* What were the arguments for the inclusion and exclusion criteria? Specifically, the county councils and the comorbities (which are also reported much more specifically in the trial registration).

-The idea was to be as pragmatic as possible and authentic to the irl clinical health care reality both patients and clinicians experience on a daily basis. Hence, we were aiming for external validity, rather than internal validity, and inclusion criterions were deliberately broad. The ambition was to describe this cohort of patients as correctly as possible, and hence their comorbidities were of interest, alongside with many other reported outcomes. The effect on all outcomes of the different care programs will be reported separately. 

* Why were the different parameters in the blood samples examined? What role do these play in the current study?

-These are important to monitor since routine blood sample parameters beyond normal reference intervals could hypothetically affect our outcomes. However, at group level, this was not the case since all routine blood test results were within normal reference intervals according to Swedish standards at Karolinska University Laboratory. We do elaborate a little on this: “Interestingly, the blood samples did not deviate from normal reference intervals indicating that patients with severe and long-lasting pain paradoxically appear “perfectly healthy”. However, more complex proteomics or pro-inflammatory markers have been shown to deviate in chronic pain patients and correlate with other health aspects such as HAD scores [54]”. The latter markers were not examined in this study.

* Why were both partial correlations and linear regression analysis examined? Did the first influence which variables were included in the latter? Or was there some other reason?

Thank you for noting this – indeed, the linear regression analyses are no longer reported in the paper due to the redundancy you describe, and we have thus removed that sentence form the statistical analysis description. Instead, we have added a description of the ANCOVA analysis result which is reported: “In addition, we used an analytic approach similar to Hassett et al [5], where the groups were divided into “high pain/low pain” and “high HAD score/low HAD score” (the group mean score was used as cut-off), and ANCOVA analyses adjusted for age were conducted with TL and TA as dependent variables.”

* Were no control variables included in the linear regression analysis, particularly gender and level of education? If so, why not? [I do want to complement the authors for controlling for multiple testing]

We do not report linear regression analyses in this paper and have removed the description form the statistical analysis section (see above). 

How were differences in age, pain duration and pain spread between the samples from the study sites handled?

*As we do not compare study sites in this study, we have not taken into account the differences in baseline variables in the study site samples in the analysis. 

* What does it mean for the study that neither TL nor TA were correlated to any of the pain measures?

-We do elaborate on this: “Another finding is the absence of correlation between TL and the pain measures selected in this study. This is surprising since others have found statistically significant correlations between TL and pain measures [5, 21-24]. In contrast, TA had a positive relationship with pain spread, i.e. high TA was associated with higher pain spread. But again, this correlation is not large enough to reach a convincing significance level with the sample size we had available. This is ambiguous since TL is negatively associated with pain spread to a notable degree”. 

The fact that TL was not linked to pain measures but rather to existential measures is something we address in the summary: “In summary, what ultimately correlates with the genomic integrity within this sample of chronic pain patients is neither quantitative pain measures or scales, nor index numbers generated for health economy calculation purposes, but instead anxiety, depression and personal, spiritual and religious beliefs”. 

Our findings hence challenge the idea that the big focus of health care on the pain itself might not be what really is important for the well being of the patients, but rather their existential situation and experiences which seem also to be linked to vital parameters such as the longevity of their DNA. 

* Why was an ANCOVA performed for the HADS and not the WHOQOL?

We agree, that it makes sense to run the same analysis and report results for WHOQOL-SRPB as well – we have now added this. 

* What theoretical considerations underlie the test for the mediation effect?

Thank you for addressing this. As we state in the article, our mediation model is intuitive, but far from definitive and that one can argue that mediation is only demonstrated according to the original Baron and Kenny approach, but not according to more recent discussions, which would require an independent interaction term to be significant. However, we nevertheless think that this potential mediation might be interesting for other researchers to scrutinize in more detail. Given the explorative design of our study we hope the reviewer agree with this.

* In the discussion section the authors discuss many findings that do not pertain to the research question. What is the purpose of this discussion?

Thank you for pointing this out. We agree that some elaborations may seem a little out of scope. However, we take the opportunity in the discussion to address gender, pain management recommendations and the global opioid crisis since they are all dimensions which the typical chronic pain patient experience with implications for their individual care programs where emotional and spiritual dimensions are lacking. 

* What is the connection of the study with opioid use (p.22-23)? This was not included in the study was it?

- We do report drug use in the result section and state in the discussion that: “despite the relation to global morbidity, mortality and doubtful long-term effectiveness, 24 % of the participants in our sample of chronic pain patients had opioids prescribed for their pain”. We think that this merits discussion of opioid use as well as the global crisis around pain management.

* Overall, I think the authors present an interesting finding, based on acceptable methodology, that certainly warrants further study. However, the background of the study in insufficiently clear and needs a more sound rationale.

- Thank you and we hope the changes and clarifications we have made have improved the manuscript to the reviewer’s satisfaction. 

Reviewer #2: Dear authors, thank you for your efforts. Your study covers a relevant topic and an important attempt to promote more holistic non-pharmacological treatments in chronic pain patients. Your methodology is sound (scales used, handling of biological materials) but the sample size is very small and relevant confounders have not been considered (tobacco and alcohol as stated by yourself, other lifestyle factors as exercise and nutrition, or patients medication specially anti-inflammatory drugs). This makes the results, specially the associations with TL/TA rather speculative and hypothetical. 

Thank you for appreciating our efforts in this challenging study and we agree that research like this is needed to find new ways to improve existential care of patients in need, such as chronic pain patients given their large suffering. 

We agree that the sample size is rather small, however given the complexity of this project with measures from cell to existential health it was not possible to enroll a larger patient basis. In the article we state that: “Because of its exploratory nature, no sample size was calculated but a minimum of 70 enrolled patients was strived for, which allows to document an effect size of r = .3 with 72% power and reaches a power of 80% with r = .32. These are the ranges of correlations that were deemed likely and clinically important”. 

We do address this as a limitation in the discussion and also state that: “The size of the correlations we found were rather small, between r = .2 and .3. Because of the small sample size and the comparatively large number of correlations, inherent in an exploratory project such as this one, we would not want to draw any definitive conclusions”. 

Hence, we agree that the findings of this exploratory project, notably the weak correlation to TL, need replication in larger clinical trials. In such future trials, the limitation of potential confounders apart from BMI, age or sex such as the one mentioned by the reviewer including tobacco or alcohol use should be considered. 

We have added the limitations suggested by the reviewer to the limitation discussion: “Indeed, levels of exercise and nutrition and patients medication should also be considered”. 

*Associations between spiritual und religious beliefs and mental health (anxiety and depression) are well documented and have been found in several studies and different contexts (pain, cardiac rehabilitation, coping with disease).

-Thank you for addressing this, and we have extended the discussion to include an extension on this: “Studies have repeatedly established positive links between religion, spirituality and health where for example religious involvement is correlated with better mental health in the areas of depression, substance abuse, and suicide (reference 1). It has also been shown that cardiovascular and cortisol stress responses of a sample of university students can be predicted by their secularity, religiosity, spirituality, and existential search (reference 2)”. 

1. Bonelli, R.M., Koenig, H.G. Mental Disorders, Religion and Spirituality 1990 to 2010: A Systematic Evidence-Based Review. J Relig Health 52, 657–673 (2013). https://doi.org/10.1007/s10943-013-9691-4

2. Schnell, T., Fuchs, D. & Hefti, R. Worldview Under Stress: Preliminary Findings on Cardiovascular and Cortisol Stress Responses Predicted by Secularity, Religiosity, Spirituality, and Existential Search. J Relig Health 59, 2969–2989 (2020). https://doi.org/10.1007/s10943-020-01008-5

6. PLOS authors have the option to publish the peer review history of their article (what does this mean?). If published, this will include your full peer review and any attached files.

Do you want your identity to be public for this peer review? For information about this choice, including consent withdrawal, please see our Privacy Policy.

Reviewer #1: Yes: Anja Visser

Reviewer #2: Yes: René Hefti M.D.

Thank you for your fine work in reviewing this manuscript and for your contributions to this research field.

Sincerely yours,

Torkel Falkenberg, on behalf of all the co-authors

---

## [Decision Letter · Decision Letter 1]

24 Oct 2023

PONE-D-23-08421R1Exploring emotional well-being, spiritual, religious and personal beliefs and telomere length in chronic pain patients - A pilot study with cross-sectional designPLOS ONE

Dear Dr. Falkenberg,

Thank you for submitting your manuscript to PLOS ONE. After careful consideration, we feel that it has merit but does not fully meet PLOS ONE’s publication criteria as it currently stands. Therefore, we invite you to submit a revised version of the manuscript that addresses the points raised during the review process.

The manuscript has been evaluated by two reviewers, and their comments are available below.

Could you please carefully revise the manuscript to address all comments raised?

We look forward to receiving your revised manuscript.

Kind regards,

Avanti Dey, PhD

Staff Editor

PLOS ONE

Journal Requirements:

Reviewers' comments:

Reviewer's Responses to Questions

**Comments to the Author**

1. If the authors have adequately addressed your comments raised in a previous round of review and you feel that this manuscript is now acceptable for publication, you may indicate that here to bypass the “Comments to the Author” section, enter your conflict of interest statement in the “Confidential to Editor” section, and submit your "Accept" recommendation.

Reviewer #1: (No Response)

Reviewer #2: (No Response)

2. Is the manuscript technically sound, and do the data support the conclusions?

Reviewer #1: Partly

Reviewer #2: Partly

3. Has the statistical analysis been performed appropriately and rigorously? 

Reviewer #1: Yes

Reviewer #2: N/A

4. Have the authors made all data underlying the findings in their manuscript fully available?

Reviewer #1: Yes

Reviewer #2: Yes

5. Is the manuscript presented in an intelligible fashion and written in standard English?

Reviewer #1: No

Reviewer #2: No

6. Review Comments to the Author

Reviewer #1: Thank you for carefully addressing the comments in your reply. Though I recognize that you have made changes in the manuscript, I would have preferred them to be a bit more elaborate to really help the reader understand why this study was conducted and is relevant and - with that - what they can learn from it. In particular, I want to ask the authors again to be more explicit about the relationship of this study to the clinical trial and the ad-hoc nature of this part of the study. I ask this mostly from a pedagogical perspective: It is helpful for readers to understand how this research process went and what struck you about the data that prompted further exploration.

On this second inspection, the results of the study seem to become less clear to me than they were. One of the reasons for this is that the authors do not provide interpretations of all the analyses displayed in the results section. It is left to the reader to interpret many the tables and draw conclusions from them. That is not helpful, given that you present this data for a reason and it had a part to play in your decision-making about analyses and your answer to the research question. Present these interpretations to the reader.

When the ANCOVA results are presented, only the analysis with the HADS is mentioned, but the results of the WHOQOL are also presented, which is confusing. Here, the reader also really needs to see how the authors interpret the different results between the partial correlations and the ANCOVA, so they can understand why the authors push on with the mediation analysis regardless of the negative results of the ANCOVA.

When the mediation analysis is presented, it is suggested that it is obvious from the results and theory that anxiety would mediate the relationship between spirituality and TL. However, I don't see an indication in the results nor in the introduction section (theoretical framework) that would suggest this. In the discussion section the authors also put this into question again. So this really needs some sentences of explanation as well. Perhaps one of my own publications can be helpful here (Visser, A., de Jager Meezenbroek, E. C., & Garssen, B. (2018). Does spirituality reduce the impact of somatic symptoms on distress in cancer patients? Cross-sectional and longitudinal findings. Social Science and Medicine, 214(August), 57–66. https://doi.org/10.1016/j.socscimed.2018.08.012), but I'm sure there are also other publications that discuss this issue. In the discussion section even mindfulness is included in the mediation hypothesis, which was not examined in the study. (it also seems the mediation effect is discussed twice in subsequent paragraphs, which is redundant)

A more minor thing that keeps puzzling me about the results is the difference between the two sites. If you didn't think they were of importance, why then did you analyze and present them? And now that you have analyzed and presented them, how do you interpret these differences and what conclusions do you draw from that in relation to your research question?

So, not withstanding my interest in this study, I still feel the authors should be more precise in their exhibition of the study and its result.

Reviewer #2: Dear authors

thank yor for revising the manuscript and adressing my comments. By having a second look at the manuscript I have make the following statements:

1. English language must be improved. The manuscript has to be revisded by a native English speaker (or a professional). I made some few comments to give you examples. But it is a general topic for the whole manuscript.

2. Please address my further comments in the attached document (see manuscript with track changes).

3. The paragraph "limitations" (ususally "strengths and limitations") must be rewritten. It is a disordered mixture of limitations and possible strengths. First mention strengths and then limitations.

7. PLOS authors have the option to publish the peer review history of their article (what does this mean?). If published, this will include your full peer review and any attached files.

Reviewer #1: **Yes: **Anja Visser

Reviewer #2: **Yes: **René Hefti MD

---

## [Author Response · Author response to Decision Letter 1]

30 Jan 2024

January 2024,

Dear Editor, 

Thank you again for considering our manuscript for publication in PLOS ONE and please find a revised version of the manuscript that addresses the points raised during the latest review process as described below, with our answers.

Reviewer #1: 

*Thank you for carefully addressing the comments in your reply. Though I recognize that you have made changes in the manuscript, I would have preferred them to be a bit more elaborate to really help the reader understand why this study was conducted and is relevant and - with that - what they can learn from it. In particular, I want to ask the authors again to be more explicit about the relationship of this study to the clinical trial and the ad-hoc nature of this part of the study. I ask this mostly from a pedagogical perspective: It is helpful for readers to understand how this research process went and what struck you about the data that prompted further exploration.

-One of our main interests lie within the field of spirituality. More specifically in the connection between psychospiritual factors and biological health. The overall aim of the prospective clinical trial was to achieve a better understanding of the possible effects of, and useful measure for evaluating, two different multimodal treatment interventions for pain. What might not have been entirely clear on clinicaltrials.gov was the explicit difference between those two care programs evaluated; one was conventional/multimodal and the other had a distinct spiritual approach. 

The aim of this study was to explore correlations of PROM’s and cellular ageing (telomere length [TL] and telomerase activity [TA]) amongst patients with chronic non-malignant pain (with some of the PROM’s relating to psychospiritual health). That is the philosophical interrelatedness between the clinical trial and this study. Moreover, as mentioned in the first rebuttal letter, we 

did not foresee the massive amount of data and realized the need for a baseline study in order to evaluate TA and TL in relation to PROMS, before adding the longitudinal perspective and evaluating the care programs. That is the pragmatic “why”.

We acknowledge that it might be uncommon to combine psychospiritual and genomic perspectives, -seemingly very different areas. However, in the context of chronic pain, these are two areas lacking research. That’s the relevance. In addition it would naturally be rather exciting if a connection between the spiritual and material dimensions could be seen. As to what can be learned, we feel that it has been sufficiently elaborated upon under “conclusion” and “implications for further studies”. In an attempt to be more explicit and to facilitate an understanding, a short introduction has been added to the very beginning:

“This study is an exploration of correlations between baseline parameters obtained from a larger, prospective clinical trial. The study spans over broad fields of knowledge, from psychospirituality to genomics. However, in the context of chronic pain, both of these fields are under-researched, which motivates the present study. In addition, this study has an explicit interest in exploring possible links between these two areas of research.”

*On this second inspection, the results of the study seem to become less clear to me than they were. One of the reasons for this is that the authors do not provide interpretations of all the analyses displayed in the results section. It is left to the reader to interpret many of the tables and draw conclusions from them. That is not helpful, given that you present this data for a reason and it had a part to play in your decision-making about analyses and your answer to the research question. Present these interpretations to the reader.

-Thank you for addressing this. Indeed, we agree that the results can be interpreted more clearly to help the reader. While we have chosen to focus the results section on the presentation of results, and placed the interpretations in the discussion section, we have carefully improved the phrasing and referring to results tables throughout the texts. Hopefully this improvement of the stringency will facilitate reader understanding.

*When the ANCOVA results are presented, only the analysis with the HADS is mentioned, but the results of the WHOQOL are also presented, which is confusing. 

-Thank you for noting this, we have now added mention of the ““highWHOQOL-SRPB/lowWHOQOL-SRPB” analysis as well. 

*Here, the reader also really needs to see how the authors interpret the different results between the partial correlations and the ANCOVA, so they can understand why the authors push on with the mediation analysis regardless of the negative results of the ANCOVA.

-It is an astute observation that these two analyses do not follow from each other. We intended them to complement each other, and we have added half a sentence to this effect at the beginning of the mediation analysis “Since the ANCOVA-analysis with dichotomized SRPB scores might have collapsed variance, and as (…)”, in addition to the following sentences on p17: “This was in order to assess whether dichotomised variables of pain, anxiety/depression, and SRPB would yield additional information about the associations with TL and TA, to what can be found through correlations with continuous variables. However, no associations were detected in this additional analysis”.

*When the mediation analysis is presented, it is suggested that it is obvious from the results and theory that anxiety would mediate the relationship between spirituality and TL. However, I don't see an indication in the results nor in the introduction section (theoretical framework) that would suggest this. In the discussion section the authors also put this into question again. So this really needs some sentences of explanation as well. Perhaps one of my own publications can be helpful here (Visser, A., de Jager Meezenbroek, E. C., & Garssen, B. (2018). Does spirituality reduce the impact of somatic symptoms on distress in cancer patients? Cross-sectional and longitudinal findings. Social Science and Medicine, 214(August), 57–66. https://doi.org/10.1016/j.socscimed.2018.08.012), but I'm sure there are also other publications that discuss this issue. In the discussion section even mindfulness is included in the mediation hypothesis, which was not examined in the study. (it also seems the mediation effect is discussed twice in subsequent paragraphs, which is redundant).

To make this connection more clear for the reader, we have added a short paragraph to the introduction (row 247-252), also quoting some references for interested readers to follow up. The new references have been added as nr 29-33 in the reference list. Since this mediation analysis is really only a serendipitous finding, we did not want to place more emphasis on it by adding still more text in the discussion. We hope that the reviewer will agree with this reasoning. 

*A more minor thing that keeps puzzling me about the results is the difference between the two sites. If you didn't think they were of importance, why then did you analyse and present them? And now that you have analysed and presented them, how do you interpret these differences and what conclusions do you draw from that in relation to your research question?

-Thank you for raising this question. We agree that it seems obsolete in relation to this study and the table has been removed. Since it is a baseline study with the same inclusion criteria for both sites, the differences are irrelevant. The data might, however, be displayed and interpreted in coming articles with longitudinal perspectives, where it may have relevance. 

*So, notwithstanding my interest in this study, I still feel the authors should be more precise in their exhibition of the study and its result.

-Well taken. We hope that our adjustments and revisions according to your suggestions has elevated and clarified the work.

Reviewer #2: 

Dear authors , thank you for revising the manuscript and addressing my comments. By having a second look at the manuscript I have to make the following statements:

1. English language must be improved. The manuscript has to be revised by a native English speaker (or a professional). I made some few comments to give you examples. But it is a general topic for the whole manuscript.

-Thank you for pointing this out. The manuscript has now been revised and grammatically corrected by a native English speaker.

2. Please address my further comments in the attached document (see manuscript with track changes)

-All the comments from the pdf file have been addressed in the word file with changes made accordingly. 

3. The paragraph "limitations" (usually "strengths and limitations") must be rewritten. It is a disordered mixture of limitations and possible strengths. First mention strengths and then limitations.

-Thank you. We have now restructured the “strengths and limitations” section and moved text from that section which was not clearly related to strengths and limitations to a new sub heading “implications for further studies”.

Many thanks for your fine work in reviewing this manuscript and also for your contributions to this research field.

Sincerely yours,

Torkel Falkenberg, on behalf of all the co-authors

---

## [Decision Letter · Decision Letter 2]

23 May 2024

PONE-D-23-08421R2Exploring emotional well-being, spiritual, religious and personal beliefs and telomere length in chronic pain patients - A pilot study with cross-sectional designPLOS ONE

Dear Dr. Falkenberg,

Thank you for submitting your manuscript to PLOS ONE. After careful consideration, we feel that it has merit but does not fully meet PLOS ONE’s publication criteria as it currently stands. Therefore, we invite you to submit a revised version of the manuscript that addresses the points raised during the review process.

Although this paper has undergone one round of revision, one of the original reviewers was no longer available, and a third reviewer was added.  This reviewer is requesting a better framing of the significance of the study in existing religion and health research, a more thorough analysis of the components of the religion/spirituality scale, and a better discussion of the significance of the findings so that the reader will have a better understanding of why telomere length should be studied in connection with religion/spirituality, and acknowledge the limitations of this small sample more openly.

We look forward to receiving your revised manuscript.

Kind regards,

Ellen L. Idler

Academic Editor

PLOS ONE

Reviewers' comments:

Reviewer's Responses to Questions

**Comments to the Author**

1. If the authors have adequately addressed your comments raised in a previous round of review and you feel that this manuscript is now acceptable for publication, you may indicate that here to bypass the “Comments to the Author” section, enter your conflict of interest statement in the “Confidential to Editor” section, and submit your "Accept" recommendation.

Reviewer #1: All comments have been addressed

Reviewer #3: (No Response)

2. Is the manuscript technically sound, and do the data support the conclusions?

Reviewer #1: Yes

Reviewer #3: Yes

3. Has the statistical analysis been performed appropriately and rigorously? 

Reviewer #1: Yes

Reviewer #3: Yes

4. Have the authors made all data underlying the findings in their manuscript fully available?

Reviewer #1: No

Reviewer #3: (No Response)

5. Is the manuscript presented in an intelligible fashion and written in standard English?

Reviewer #1: Yes

Reviewer #3: Yes

6. Review Comments to the Author

Reviewer #1: Thank for carefully addressing all of the comments. The rationale of the study is now sufficiently clear.

Reviewer #3: The findings from this pilot study are very interesting. While the paper has improved from the initial submission, I have several concerns that the authors should consider.

1) I do not think the introduction/background adequately justifies the current study. What is the motivation beyond these topics not being previously investigated? Why might religion/ spirituality impact telomere length among those with chronic pain? Why might religion/ spirituality be mediated by mental health in this population? Despite space limitations, some elaboration is necessary.

I suggest reviewing the following for a general framework:

Hill, T. D., Ellison, C. G., Burdette, A. M., Taylor, J., & Friedman, K. L. (2016). Dimensions of religious involvement and leukocyte telomere length. Social Science & Medicine, 163, 168-175.

Page, R. L., Peltzer, J. N., Burdette, A. M., & Hill, T. D. (2020). Religiosity and health: A holistic biopsychosocial perspective. Journal of holistic nursing, 38(1), 89-101.

Handbook of Religion and Health- Koenig, VanderWeele, Peteet 2024 (There is a section on religion and biological functioning).

I also suggest reviewing the following for information on how religion may work among those with health problems:

Idler, E. L., & Kasl, S. V. (1992). Religion, disability, depression, and the timing of death. American journal of Sociology, 97(4), 1052-1079.

Hill, T. D., Burdette, A. M., Taylor, J., & Angel, J. L. (2016). Religious attendance and the mobility trajectories of older Mexican Americans: An application of the growth mixture model. Journal of Health and Social Behavior, 57(1), 118-134.

2) Did the authors conduct additional analyses to exam individual aspects of the religion/ spirituality scale with the focal variables? Given that the most significant findings center around this scale, additional analyses may be insightful.

2) I also suggest strengthening the discussion by elaborating on what we can take from these findings. The current discussion section is somewhat vague and would benefit from examining previous research on religion/spirituality and biological functioning.

7. PLOS authors have the option to publish the peer review history of their article (what does this mean?). If published, this will include your full peer review and any attached files.

Reviewer #1: **Yes: **Anja Visser

Reviewer #3: No

---

## [Author Response · Author response to Decision Letter 2]

28 Jul 2024

Dear Editor, 

Thank you once again for considering our manuscript for publication in PLOS ONE and please find a revised version of the manuscript that addresses the points raised during this latest review process as described below, with our answers in bold.

Reviewer #1: 

* Thank for carefully addressing all of the comments. The rationale of the study is now sufficiently clear.

-Thank you for the reviewing and help with elevating our work

Reviewer #3: 

The findings from this pilot study are very interesting. While the paper has improved from the initial submission, I have several concerns that the authors should consider.

1) I do not think the introduction/background adequately justifies the current study. What is the motivation beyond these topics not being previously investigated? Why might religion/ spirituality impact telomere length among those with chronic pain? Why might religion/ spirituality be mediated by mental health in this population? Despite space limitations, some elaboration is necessary.

I suggest reviewing the following for a general framework:

Hill, T. D., Ellison, C. G., Burdette, A. M., Taylor, J., & Friedman, K. L. (2016). Dimensions of religious involvement and leukocyte telomere length. Social Science & Medicine, 163, 168-175.

Page, R. L., Peltzer, J. N., Burdette, A. M., & Hill, T. D. (2020). Religiosity and health: A holistic biopsychosocial perspective. Journal of holistic nursing, 38(1), 89-101.

Handbook of Religion and Health- Koenig, VanderWeele, Peteet 2024 (There is a section on religion and biological functioning).

I also suggest reviewing the following for information on how religion may work among those with health problems:

Idler, E. L., & Kasl, S. V. (1992). Religion, disability, depression, and the timing of death. American journal of Sociology, 97(4), 1052-1079.

Hill, T. D., Burdette, A. M., Taylor, J., & Angel, J. L. (2016). Religious attendance and the mobility trajectories of older Mexican Americans: An application of the growth mixture model. Journal of Health and Social Behavior, 57(1), 118-134.

-We are grateful for this advice. We have reviewed the studies indicated by the reviewers and included four of them plus an additional one that may help clarify the definitional problems. We added two paragraphs to the MS (lines 122 to 140). We hope that this clarifies the purpose better. Also, it should be noted that our research question was rather broad and because of some ethical concerns about the load of questions to be given to the patients we were unable to implement a more differentiated measurement approach to spirituality and religiosity.

2) Did the authors conduct additional analyses to exam individual aspects of the religion/ spirituality scale with the focal variables? Given that the most significant findings center around this scale, additional analyses may be insightful.

-No, we did not. We already added one additional analysis, the mediation model, to those stipulated by our protocol and we mentioned this. Since already this additional analysis met with some concerns by other reviewers in previous rounds of reviews we thought it not advisable to go into further details. Apart from that, we had no further a-priori hypotheses about specific hypotheses of how aspects of mindfulness or of the spiritual well being module of the WHOQol would be related to our potential outcome variables. As this was only one aspect of our study, we did not feel justified to further explore the data set. Exploration of such data always comes with the danger of capitalization of chance, especially if no a-priori hypotheses have been formulated. Hence, we would rather not explore this data-set further but rather develop a new study with more focused hypotheses.

3) I also suggest strengthening the discussion by elaborating on what we can take from these findings. The current discussion section is somewhat vague and would benefit from examining previous research on religion/spirituality and biological functioning.

-Thank you for this suggestion. We have added a paragraph where we added some previous findings and strengthened the discussion and conclusions accordingly. We hope, this makes the manuscript acceptable.

Many thanks for your fine work in reviewing this manuscript and also for your contributions to this research field.

Sincerely yours,

Torkel Falkenberg, on behalf of all the co-authors

---

## [Editor Report · Decision Letter 3]

2 Aug 2024

Exploring emotional well-being, spiritual, religious and personal beliefs and telomere length in chronic pain patients - A pilot study with cross-sectional design

PONE-D-23-08421R3

Dear Dr. Falkenberg,

We’re pleased to inform you that your manuscript has been judged scientifically suitable for publication and will be formally accepted for publication once it meets all outstanding technical requirements.

Kind regards,

Ellen L. Idler

Academic Editor

PLOS ONE
---

## [Editor Report · Acceptance letter]

13 Aug 2024

PONE-D-23-08421R3 

PLOS ONE

Dear Dr. Rönne-Petersén, 

I'm pleased to inform you that your manuscript has been deemed suitable for publication in PLOS ONE. Congratulations! Your manuscript is now being handed over to our production team.

Kind regards, 

on behalf of

Professor Ellen L. Idler 

Academic Editor

PLOS ONE